# Mapping the Mind: Gray Matter Signatures of Personality Pathology in Female Adolescent Anorexia Nervosa Persist Through Treatment

**DOI:** 10.3390/jcm14155438

**Published:** 2025-08-01

**Authors:** Lukas Lenhart, Manuela Gander, Ruth Steiger, Agnieszka Dabkowska-Mika, Malik Galijasevic, Stephanie Mangesius, Martin Fuchs, Kathrin Sevecke, Elke R. Gizewski

**Affiliations:** 1Department of Radiology, Medical University of Innsbruck, Anichstrasse 35, A-6020 Innsbruck, Austria; lukas.lenhart@i-med.ac.at (L.L.); ruth.steiger@i-med.ac.at (R.S.); agnieszka.dabkowska-mika@i-med.ac.at (A.D.-M.); malik.galijasevic@i-med.ac.at (M.G.); stephanie.mangesius@i-med.ac.at (S.M.); elke.gizewski@i-med.ac.at (E.R.G.); 2Neuroimaging Research Core Facility, Medical University of Innsbruck, Anichstrasse 35, A-6020 Innsbruck, Austria; 3Department of Child and Adolescent Psychiatry, Medical University of Innsbruck, Anichstrasse 35, A-6020 Innsbruck, Austria; martin.fuchs@tirol-kliniken.at (M.F.); kathrin.sevecke@i-med.ac.at (K.S.); 4Institute of Psychology, Leopold-Franzens University of Innsbruck, Universitätsstrasse 3-5, A-6020 Innsbruck, Austria

**Keywords:** anorexia nervosa, gray matter, personality disorder, adolescence, voxel-based morphometry

## Abstract

**Background**: Comorbid personality disorders (PDs) in patients with anorexia nervosa (AN) are associated with increased psychopathology, higher suicide risk, and poorer treatment response and outcomes. This study aimed to examine associations between gray matter (GM) volume and PDs in female adolescents with AN before and after short-term psychotherapeutic and nutritional therapy. **Methods**: Eighteen female adolescents with acute AN, mean age 15.9 years, underwent 3T magnetic resonance imaging before and after weight restoration. The average interval between scans was 2.6 months. Structural brain changes were analyzed using voxel-based morphometry. PDs were assessed using the Structured Clinical Interview for DSM-IV Axis II Disorders (SCID II) and the Assessment of Identity Development Questionnaire. **Results**: SCID-II total scores showed significant positive associations with GM volume in the mid-cingulate cortex at both time points and in the left superior parietal–occipital lobule at baseline. The histrionic subscale correlated with GM volume in the thalamus bilaterally and the left superior parietal–occipital lobule in both assessments, as well as with the mid-cingulate cortex at follow-up. Borderline and antisocial subscales were associated with GM volume in the thalamus bilaterally at baseline and in the right mid-cingulate cortex at follow-up. **Conclusions**: PDs in female adolescent patients with AN may be specifically related to GM alterations in the thalamus, cingulate, and parieto-occipital regions, which are present during acute illness and persist after weight restoration therapy.

## 1. Introduction

Personality disorders (PDs) are severe, enduring patterns of behavior and cognition that complicate psychiatric treatment, particularly in anorexia nervosa (AN) [1,2]. Anorexia nervosa, as defined by the DSM-5, is an eating disorder marked by a persistent restriction of food intake leading to significantly low body weight, an intense fear of gaining weight or becoming fat, and a distorted perception of body image or denial of the seriousness of low body weight. It includes two subtypes: the restricting type, where weight loss is achieved through dieting or excessive exercise, and the binge-eating/purging type, involving recurrent episodes of bingeing or purging behaviors such as vomiting or misuse of laxatives [1,2].

Comorbid PDs in AN are linked to heightened psychopathology, suicide risk, reduced treatment response, and poorer outcomes in adolescents and adults [3,4]. Avoidant (53%) and obsessive–compulsive PDs (33%) are most prevalent in restrictive-type AN [5]. These comorbidities worsen clinical challenges, including fragile therapeutic alliances, premature treatment discontinuation, chronicity, and lower symptom improvement [4,6,7]. Neurobiological insights into PDs in adolescent AN remain limited. Structural neuroimaging studies suggest PD-specific gray matter (GM) alterations. For instance, borderline PD—common in AN (30% prevalence; [8])—is associated with increased GM volumes in the cingulate cortex, precuneus, prefrontal, and parietal regions [9,10], alongside reductions in the orbitofrontal gyrus and anterior cingulate cortex [11]. Obsessive–compulsive PD correlates with larger thalamic volumes and smaller orbitofrontal cortices [12,13], while avoidant PD traits (e.g., social anxiety) are linked to diffuse cortical GM enlargement and subcortical reductions [14]. Such abnormalities may underpin emotional dysregulation and poor treatment responses. Symptomatic overlaps between PDs and AN behaviors suggest shared neurobiological mechanisms. Obsessive–compulsive PD traits like rigid perfectionism mirror AN’s meticulous dietary rules [15], while borderline PD’s impulsivity aligns with AN’s self-destructive behaviors [16]. Despite these parallels, few studies explore GM correlates of PDs in adolescent AN, a critical gap given PDs’ role in treatment resistance.

Adolescents with anorexia nervosa (AN) in the acute phase consistently exhibit widespread reductions in gray matter (GM) volume, particularly in cortical and subcortical regions [17,18]. These changes are often interpreted as a direct consequence of malnutrition [19,20]. Notably, rapid partial normalization of GM volume after short-term weight restoration has been reported [21], supporting the view that at least some structural deficits may be reversible. However, the extent and completeness of this recovery remain controversial. Seitz et al. (2014 [22]) conducted a meta-analysis indicating that although some structural recovery is possible, full normalization—especially in adolescents—is not always observed, even after prolonged remission. Recent work by Stratton et al. (2024 [23]) further challenges the notion of full reversibility, demonstrating transient brain aging signatures in adolescents that may persist beyond nutritional rehabilitation. Similarly, further data suggest dynamic but incomplete recovery patterns [24], pointing toward a complex interplay between neurodevelopment and illness duration. Psychological factors such as symptom severity and attachment trauma also appear to modulate neurostructural outcomes. While some studies link greater symptom severity with less GM recovery [24], others find no consistent associations, suggesting a controversial and multifactorial influence [25]. In addition, a history of attachment trauma has been associated with persistent GM alterations and disrupted frontolimbic connectivity in adolescents with AN [26], reinforcing the notion that brain changes in AN are not purely nutritional but shaped by deeper psychological vulnerabilities. This study investigates associations between PD severity and GM volumes using voxel-based morphometry (VBM) in adolescents with AN pre and post treatment. VBM is particularly well-suited for this study because it enables unbiased, automated, whole-brain analysis of GM differences without requiring a priori selection of regions, which is crucial when exploring neurostructural correlates of PD severity and its clinical manifestations in AN. To date, no study has systematically examined the influence of personality disorders on gray matter changes in anorexia nervosa, despite their high prevalence in clinical populations and their established relevance for treatment outcomes [27]. We hypothesize that PDs in AN are tied to prefrontal, temporal, limbic, and cingulate GM alterations, which persist after weight restoration. Aligning with dimensional PD models (ICD-11; DSM-5 AMPD), we assess PD severity as a continuum rather than using categorical diagnoses, which is increasingly supported in adolescent populations [28]. Although including a non-PD control group could enhance group comparisons, dimensional approaches allow for a more developmentally sensitive and clinically informative analysis of personality dysfunction across the full spectrum. Dimensional models better predict functioning and outcomes [3] yet remain unexplored in GM recovery studies of AN [29,30].

We address two aims: (1) correlations between PD severity and clinical symptomatology and (2) longitudinal GM-PD associations pre and post treatment. This work advances understanding of neurostructural markers underlying PDs in AN, potentially informing targeted interventions for this high-risk population.

## 2. Materials and Methods

### 2.1. Participants

Female adolescent AN patients (age range from 14 to 18 years) were recruited from an in-house database at the Department of Child and Adolescent Psychiatry, Psychotherapy and Psychosomatics. Restrictive-type AN and comorbid PDs were diagnosed based on the Structured Clinical Interview for DSM-IV (SCID-I and II [31]). Extremely underweight AN patients (BMI for age <5th percentile) who required pediatric treatment for medical stability and improvement in cognitive functioning prior to the treatment were not eligible for study participation. Out of 27 female inpatients, patients were excluded due to movement artifacts during the MRI scan (n = 2) and loss of neuropsychological (n = 3) or MRI follow-up (n = 6). The final patient sample consisted of n = 18 female AN patients (mean age: 15.9 ± 1.2 years). All adolescents were classified with restrictive-type AN and had an intelligence score within the normal range (>84 in the Hamburg Wechsler Intelligence Scale [32]), a sufficient knowledge of the German language for psychological testing, no acute or chronic functional or somatic diseases (i.e., tumors, strokes, heart conditions, previous head trauma), no substance abuse in the past, and no other contraindications for MRI.

Patients were recruited from 2015 to 2017 within the first week of admission. In clinical routine, the diagnosis of a comorbid PD is consistently communicated to both the patients and their parents and is also shared and discussed within the multidisciplinary treatment team. This team-based approach ensures that care is coordinated across psychiatric, psychotherapeutic, nursing, and social work professionals, providing a comprehensive and individualized treatment plan. Treatment included psychoeducation, psychotherapy, and family therapy. Nutritional therapy included a daily calorie intake of 2400 to 3000 kcal/day, which was monitored daily. In our clinic, adolescents with AN receive structured nutritional therapy aimed at restoring physical health and normalizing eating behavior. This includes an individualized meal plan based on age-appropriate needs, supervised meals to ensure compliance, and regular weight monitoring. Psychoeducation and family involvement are essential components, and the approach is closely coordinated with the multidisciplinary treatment team. Since all participants were between 14 and 18 years of age, BMI was classified using WHO BMI-for-age percentiles for 5- to 19-year-olds, where significantly low weight is defined as a BMI for age below the 5th percentile [33]. In this context, stable weight recovery was defined as reaching a BMI for age at or above the 5th percentile, indicating sufficient weight gain for the given age and developmental stage.

After achieving medical stability, the first MRI scan (time 1, tp1; n = 18) and a psychological assessment were performed. The second MRI scan (time point 2, tp2; n = 18) was performed after weight restoration (BMI for age ≥5th percentile). The time between MRI scans averaged 2.6 ± 0.9 months. The weight assessment was conducted prior to the initial MRI scan and following the final MRI scan during the patient’s stay at the clinic.

The study was approved by the local ethics committee (AN2015-0036) and conducted in accordance with the Declaration of Helsinki. We obtained informed consent from all participants and their parents/legal guardians. While the sample used partially overlaps with samples reported in two previous publications [26,34], the current investigation focuses specifically on the concept of personality disorders. As such, the research question, hypotheses, methodological approach, and conclusions of this study substantially differ from those of the earlier works.

### 2.2. Measures

#### 2.2.1. Eating Disorder Diagnosis and Symptomatology

We diagnosed anorexia nervosa (restrictive type) based on the Structured Clinical Interview for DSM-IV (SCID-I, German translation: [31]). The interview was conducted by trained clinical psychologists at our clinic. Research on psychometric properties demonstrates good data validity and reliability of the SCID in determining the accuracy of DSM diagnoses in adult and adolescent age groups [31,35] with Kappa values above 0.70 for eating disorders [35].

To further assess the severity of eating disorder symptoms, we used the German version of the Eating Disorder Inventory-2 [36]. The EDI-2 questionnaire consists of 91 items with a 6-point Likert scale and 11 subscales that can be added up to a sum score: bulimia, body dissatisfaction, drive for thinness, ineffectiveness, interpersonal distrust, perfectionism, interoceptive awareness, impulse regulation, maturity fears, asceticism (provisional), and social insecurity (provisional).

#### 2.2.2. Personality Disorder Assessment

Trained clinical psychologists at our clinic conducted the SCID-II to assess the 10 personality disorder diagnoses in our adolescent sample: avoidant, obsessive–compulsive, antisocial, borderline, paranoid, histrionic, dependent, schizotypal, schizoid, and narcissistic. In accordance with the new dimensional classification models of the ICD-11 and the DSM-5 Alternative Model of PD (AMPD), we adopted a dimensional concept of PDs, which provides a more accurate picture regarding the patient’s global PD severity during treatment [29,30]. This dimensional PD profile allows quantification of PD severity along one dimension [30]. This approach acknowledges that personality pathology in youth often manifests along a continuum of severity, making dimensional assessment more developmentally sensitive and clinically informative than rigid categorical diagnoses. Therefore, rather than subdividing the sample into PD and non-PD groups, we aimed to capture variability across the spectrum of personality dysfunction. Our dimensional framework allows for a nuanced investigation of the association between severity of personality dysfunction and brain structural alterations, which categorical designs may not adequately detect.

Depending on the level of psychopathology, the procedure usually takes between one and two hours. Although the SCID-II is often used in adult samples, it has been administered successfully in younger age groups as well [31]. As previously mentioned, the reliability and validity of the data of the SCID-I and II can be considered as good. Furthermore, numerous cross-national epidemiologic studies were conducted to demonstrate reliability and validity of the SCID-II in non-English-speaking samples. The results demonstrate superior validity over other standard clinical interviews during the intake episode [37].

For additional dimensional assessment of PDs, we used the Assessment of Identity Development in Adolescence (AIDA, [38]). This self-report questionnaire assesses on a 5-point Likert scale (0 = strong disagreement to 4 = strong agreement) an individual’s identity development ranging from identity integration to identity diffusion. Higher scores indicate higher levels of personality dysfunction on the criterion identity according to the ICD-11 dimensional classification of PDs. In several studies, the AIDA demonstrated good validity and reliability scores (i.e., Cronbach’s alpha = 0.95 for internal consistency, effect sizes from 1.04 to 2.56 for criterion validity) in adolescent samples [39].

#### 2.2.3. Childhood Trauma

We used the childhood trauma questionnaire to evaluate subjectively reported traumatic childhood experiences. The 28-item questionnaire retrospectively assesses emotional, physical, and sexual abuse and emotional and physical neglect on a 5-point Likert scale from never true to very often true (0–5). The global maltreatment scale ranges from 25 to 125, with higher total scores indicating a higher severity of maltreatment experiences in childhood. This instrument is an internationally accepted and widely used questionnaire that demonstrates good construct validity and reliability and validity in large adolescent and adult samples (n = 2000) (i.e., acceptable internal consistency of Cronbach’s alpha ≥ 0.80 and good construct validity with positive correlations with depression, r = 0.36 (*p* < 0.001), and with anxiety, r = 0.40, *p* < 0.001, and negative correlations with life satisfaction, r = −0.23, *p* < 0.001) [40].

#### 2.2.4. MRI Acquisition and Processing

All participants were scanned using a 3.0 Tesla MRI system (Siemens Verio, Erlangen, Germany) following a standardized imaging protocol. The coronal T1-weighted 3D MPRAGE sequence included a repetition time (TR) of 1950 ms, an echo time (TE) of 3.3 ms, a flip angle of 9°, an in-plane field of view of 220 × 178 mm, and a slice thickness of 1 mm. This resulted in 160 contiguous transversal slices with a voxel resolution of 0.9 × 0.7 × 1 mm. Experienced neuroradiologists reviewed all scans to rule out subclinical abnormalities, and each dataset was visually inspected for motion artifacts or instrument-related issues. Only scans that passed these quality control checks, as well as the homogeneity assessment implemented in the Computational Anatomy Toolbox (CAT12; Structural Brain Mapping group, University of Jena, Jena, Germany), were included in further analysis.

For whole-brain analysis, MRI data were processed using the CAT12 within the SPM12 environment (Statistical Parametric Mapping, Institute of Neurology, London, UK [41,42]), running on MATLAB R2018b. The preprocessing pipeline involved several key steps: bias-field correction, skull stripping, and alignment to the Montreal Neurological Institute (MNI-152) standard space. Images were then segmented into gray matter, white matter, and cerebrospinal fluid compartments [43]. Spatial normalization was performed using the DARTEL algorithm [44], after which the segmented tissue maps were smoothed with a Gaussian kernel of 4 × 4 × 4 mm full width at half maximum (FWHM). To minimize signal noise, a masking threshold of 10% was applied.

This automated processing workflow incorporated multiple levels of quality assurance, including both manual and automated checks at each stage. The use of standardized pipelines and established neuroimaging software ensures reproducibility and reliability in the analysis of structural brain data. Such robust, modular pipelines are common in contemporary neuroimaging research, where they facilitate the efficient and consistent processing of large datasets while allowing for comprehensive quality control and flexible adaptation to specific research needs [45].

### 2.3. Statistical Analysis

To examine the relationships among variables from the SCID, CTQ, AIDA, and EDI, Spearman rank-order correlations were conducted using SPSS version 25 (SPSS Inc., Chicago, IL, USA). For whole-brain voxel-based analyses, a general linear model was employed to compare cross-sectional and longitudinal data between baseline and follow-up time points. This analysis utilized a flexible factorial design implemented in SPM12 to assess the association between GM volume and SCID scores. In all statistical models, TIV and age were treated as nuisance variables, and BMI was additionally included as a covariate to account for weight-related variance. Statistical significance was set at *p* < 0.01, with correction for multiple comparisons via the family-wise error (FWE) rate at *p* < 0.05.

## 3. Results

### 3.1. Neuropsychological Background Tasks

All participants’ scores were in the average range in the SCID battery (avoidant, dependent, obsessive–compulsive, paranoid, schizotypal, schizoid, histrionic, narcissistic, and borderline) (see Table 1). From a total of 18 adolescent patients diagnosed with AN, n = 7 (39%) patients were classified with a PD (avoidant PD n = 3, obsessive–compulsive PD n = 3, borderline PD n = 1). This rate is broadly similar to the findings of Magallón-Neri et al. [5], who reported a prevalence of 33% in a larger clinical sample. It should be noted that our sample consists of inpatients with more severe psychopathology, which may explain the slightly higher rate. Performance in the SCID (total score) correlated with the AIDA (r = 0.534, *p* = 0.023) and EDI (r = 0.709, *p* = 0.002). No significant association was found between the SCID and CTQ scores.

### 3.2. Voxel-Based Correlations Between Morphometric and Psychometric Data

Significant associations between SCID-II scores and GM volume were found in the group of 18 patients with acute AN. At baseline (time point 1), SCID-II total scores showed significant positive correlations with GM in the mid-cingulate (MNI: 5, 18, 36; t = 5.2, *p* = 0.003) and the left superior parieto-occipital region (MNI: −26, −69, 39; t = 6.3, *p* = 0.005), both corrected for multiple comparisons at the cluster level (family-wise error rate, FWE) (Table 2).

For specific PD traits, distinct patterns emerged. Histrionic traits were associated with GM in the thalamus (MNI: −17, −26, 5; t = 6.5, *p* = 0.002; MNI: 15, −20, 9; t = 4.7, *p* = 0.009) and in the left superior parieto-occipital region (MNI: −30, −66, 54; t = 5.9, *p* = 0.001). Borderline traits were linked to GM in the thalamus (MNI: 0, −21, 9; t = 4.8, *p* < 0.001), and antisocial traits also showed GM correlations in the thalamus (MNI: −9, −21, 5; t = 5.1, *p* < 0.001).

At follow-up (time point 2), SCID-II total scores again correlated with GM in the mid-cingulate (MNI: 6, −18, 48; t = 6.1, *p* < 0.001) and the left superior parieto-occipital region (MNI: −27, −60, 48; t = 4.2, *p* = 0.010). For histrionic traits, GM correlations were observed in the mid-cingulate (MNI: 3, −18, 44; t = 8.8, *p* < 0.001), thalamus (MNI: −5, −11, 15; t = 7.1, *p* = 0.001), left superior parieto-occipital region (MNI: −30, −68, 54; t = 7.7, *p* < 0.001), and right angular gyrus (MNI: 42, −74, 36; t = 6.2, *p* = 0.001). Borderline traits were associated with GM in the right mid-cingulate (MNI: 14, −21, 45; t = 5.4, *p* = 0.027), and antisocial traits showed a correlation in the right mid-cingulate as well (MNI: 5, −20, 42; t = 4.4, *p* = 0.05) (Figure 1 and Figure 2).

All reported associations survived correction for multiple comparisons at the cluster level (FWE; *p* < 0.05 or as indicated), with a height threshold of *p* < 0.01. These findings highlight specific brain regions where GM volume was associated with the presence and severity of PD features in patients with acute AN at both baseline and follow-up. No statistically significant changes were observed across the longitudinal assessment of GM corrected for BMI.

## 4. Discussion

This study provides evidence that PDs in adolescents with AN are associated with specific GM regions. PD severity, assessed by the SCID-II, significantly correlated with identity diffusion (AIDA) and eating disorder symptom severity (EDI) but not with childhood trauma (CTQ). VBM revealed significant associations with the mid-cingulate, thalamus, and left superior parieto-occipital gyrus in the acute phase, persisting after weight recovery [9].

PDs were significantly associated with GM volume in the mid-cingulate, consistent with findings of increased GM volume in this region in young adults with borderline PD [9]. The mid-cingulate’s involvement could indicate pain, anxiety, and deficits in the emotional–cognitive interaction in female adolescent AN patients with PD. This aligns with studies showing increased mid-cingulate activation in borderline PD patients in response to emotional pain and abandonment themes [46,47]. Significant correlations were found between PDs and GM volume in the left superior parietal lobule. This region, along with the precuneus, is involved in body representation, conscious information processing, and self-reflection [48,49]. Increased volume might reflect disturbances in self-referential and emotional processing, potentially linked to dissociative symptoms in borderline PD [9,50]. Notably, these structural brain features persisted after weight restoration, contrasting with previous findings that most region-specific abnormalities in AN normalize during nutritional therapy [18,51]. This suggests that PD pathology might represent enduring abnormalities in GM plasticity, persisting after treatment [26,34].

The histrionic, borderline, and antisocial SCID dimensional subscales were associated with higher thalamus volume, particularly at baseline. This aligns with previous studies demonstrating structural changes in the thalamus in borderline PD patients [52,53]. The thalamus’s role in memory, consciousness, emotion, and reward suggests that GM changes in this region might be associated with emotional dysregulation and impulsivity in patients with borderline symptoms [52]. Unexpectedly, no significant correlations were found between obsessive–compulsive and avoidant PDs and GM volume of the thalamus, contrary to previous findings in obsessive–compulsive PD and social anxiety disorder [14,54,55].

The lack of significant longitudinal GM differences in the present study may be attributed to several factors. PDs are highly heterogeneous in their presentation, with substantial variability in symptoms, severity, and course among affected individuals [56]. This heterogeneity complicates the detection of consistent GM longitudinal patterns, particularly when sample sizes are limited, increasing the likelihood that subtle or moderate changes over time remain undetected. Previous research has demonstrated that brain structural changes in AN are strongly influenced by acute and chronic weight loss, as well as by fluctuations in BMI [26]. Given that brain volume alterations in AN are often weight-dependent and that weight status can fluctuate during the illness’s course, it is plausible that any longitudinal associations between PD features and GM volume were overshadowed by the more pronounced effects of weight changes. This could further explain why, despite baseline associations, no significant longitudinal differences were observed in this sample. Additionally, PDs are typically characterized by consistent patterns of maladaptive thoughts, emotions, and behaviors that persist over an extended period of time [28]. Due to their relative temporal stability, significant improvements are generally not expected within short observation periods such as the 2.5-month interval used in this study. This further supports the interpretation that, despite baseline associations, no significant longitudinal changes were observed in this sample when corrected for BMI.

Our findings reveal that specific structural brain alterations in adolescents with acute AN persist even after nutritional restructuring. Despite improvements in BMI over the course of treatment, no statistically significant changes were observed in GM volume when corrected for BMI, suggesting that certain neuroanatomical differences may represent stable trait-related features rather than solely consequences of malnutrition [19]. This persistence highlights the importance of the enduring neurobiological vulnerabilities underlying AN, which may contribute to disease chronicity and influence treatment response [57]. These results also suggest the necessity to differentiate brain abnormalities based not only on BMI or eating disorder symptoms but also on predisposing factors like comorbid PDs. Recognizing characteristics of PDs–such as emotional dysregulation, interpersonal problem behaviors, or impulsivity–may help clinicians to tailor therapeutic strategies more effectively. For instance, individuals with higher symptoms of borderline or histrionic pathology may benefit from interventions targeting affect regulation and relational dynamics. Moreover, these neurostructural patterns may act as early markers of poorer treatment response or relapse risk, supporting the use of dimensional personality assessments in early diagnostics. Integrating disorder-specific interventions for PDs alongside nutritional and psychiatric care might enhance long-term outcomes by addressing both the psychological and neurobiological dimensions of AN recovery. Understanding these lasting brain changes is therefore crucial for developing comprehensive therapeutic approaches that address not only weight restoration but also the underlying neural substrates associated with PDs and symptom maintenance [58].

The study’s limitations include a relatively small sample size and the focus exclusively on adolescent female inpatients. While this sample reflects the demographic most affected by AN and enhances internal validity, it limits the generalizability of the findings to male patients and older age groups. Increasing evidence from structural and functional neuroimaging studies suggests that AN may manifest differently across sexes. For instance, Leehr et al. (2019 [59]) reported sex-specific associations between genetic risk for AN and prefrontal brain structure, indicating potentially distinct neurobiological mechanisms. Furthermore, Zhang et al. [60] demonstrated that alterations in the amygdala and hippocampal subfields were particularly pronounced in female patients, highlighting the relevance of gender-specific analyses in structural brain research. While our study was designed to examine a homogeneous clinical group, future research should aim to include male participants and consider developmental stage more explicitly, to clarify whether the observed structural alterations are specific to adolescent females or reflect broader patterns within the AN population. Furthermore, we assessed a short-term weight restoration period and conducted an exclusive analysis of GM volume. The dimensional assessment of PDs, while aligning with current DSM-5 AMPD and ICD-11 approaches, may not capture extreme levels of symptoms that could reflect stronger structural brain abnormalities. While stable weight recovery was defined as achieving a BMI for age at or above the 5th percentile, this cutoff may not fully capture individual differences in nutritional status, as it does not account for pre-morbid weight trajectories or physiological markers of malnutrition.

Recent research increasingly supports the validity and clinical utility of diagnosing personality disorders during adolescence. Empirical findings indicate that personality pathology can be identified reliably in individuals under 18 and that early recognition allows for more targeted interventions and may help prevent long-term impairments in functioning, including academic, social, and emotional outcomes. Diagnostic models such as the DSM-5 Alternative Model and ICD-11 now acknowledge the developmental appropriateness of assessing personality functioning in youth [23]. At the same time, in clinical populations marked by significant malnutrition, such as adolescents with AN, it is important to consider how nutritional status may influence personality-related domains—particularly self-functioning and interpersonal relationships. As shown by Beadle et al. [61], malnutrition in anorexia nervosa can impair social cognition and interpersonal functioning, potentially mimicking or exacerbating personality pathology. This raises the concern that current impairments may reflect a state effect rather than enduring traits. To disentangle these influences, longitudinal assessment of personality functioning—before, during, and after nutritional rehabilitation—would be crucial to clarify whether observed deficits persist beyond the acute phase and thus reflect true personality pathology.

This study used VBM to analyze GM alterations in adolescents with AN, leveraging its strength in detecting region-specific volumetric differences. While graph theory-based network approaches provide insights into brain-wide connectivity [62], VBM remains optimal for hypothesis-driven studies of localized structural changes, particularly given the absence of multimodal imaging data (e.g., resting-state fMRI/DTI) required for advanced network analyses here. Future work should combine these methods to explore PD influence on broader network dynamics. Extending the conclusions drawn by Fortea et al. [63], who demonstrate that disregarding comorbidities may obscure disorder-specific neural correlates, our results further indicate that the presence of personality disorders is a critical confounding and contributing factor in the neurobiology of anorexia nervosa. This underscores the clinical and research imperative to develop co-occurrence-sensitive neuroimaging biomarkers for improved diagnostic specificity and treatment personalization in eating disorders.

## 5. Conclusions

Our findings support the hypothesis that PDs are associated with specific focal GM alterations in brain regions important for AN symptomatology and potentially related to treatment outcomes in adolescent patients. In particular, PDs in female adolescent patients with AN may be specifically related to GM alterations in the thalamus, cingulate, and parieto-occipital regions, which are present during acute illness and persist after weight restoration therapy.

## Figures and Tables

**Figure 1 jcm-14-05438-f001:**
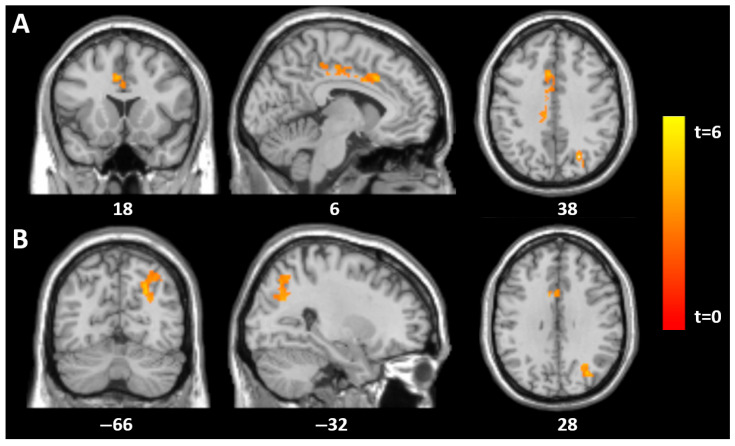
Statistical parametric mapping (t) intensity projection maps rendered onto a stereotactically normalized MRI scan: voxel cluster of significant associations between the SCID (total score) and GM volume in the mid-cingulate (**A**) and the left superior parieto-occipital gyrus (**B**) in 18 AN patients at tp1 (statistical significance is thresholded at *p* < 0.01, FWE *p* < 0.05 corrected at the cluster level). The number at the bottom of each MRI scan corresponds to the z coordinate in MNI space.

**Figure 2 jcm-14-05438-f002:**
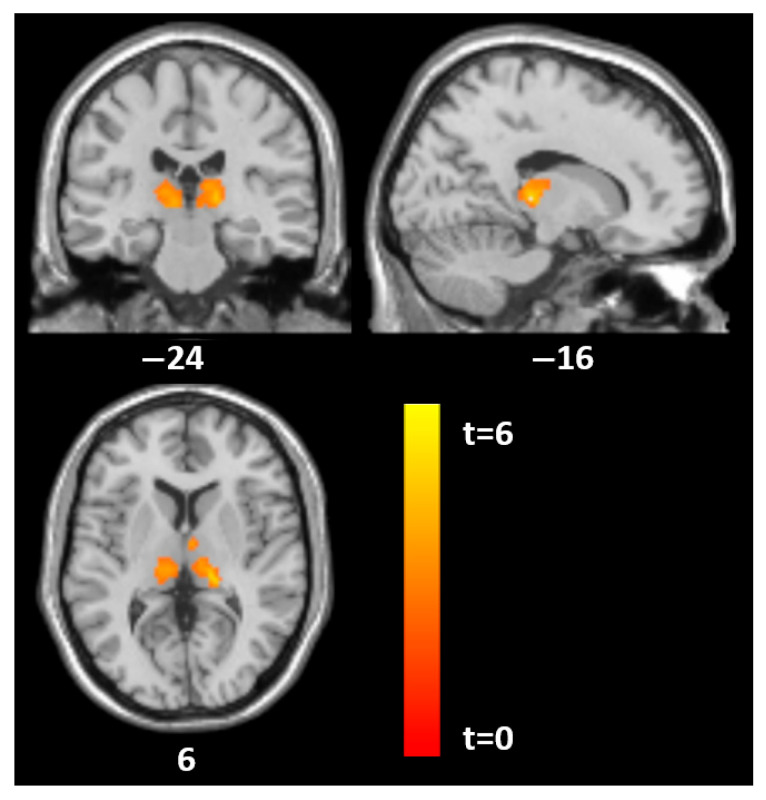
Statistical parametric mapping (t) intensity projection maps rendered onto a stereotactically normalized MRI scan: voxel cluster of significant associations between the SCID histrionic subscale and GM volume in the thalamus on both sides in 18 AN patients at tp1 (statistical significance is thresholded at *p* < 0.01, FWE *p* < 0.05 corrected at the cluster level). The number at the bottom of each MRI scan corresponds to the z coordinate in MNI space.

**Table 1 jcm-14-05438-t001:** Psychological assessment of PD and ED severity and childhood trauma.

Task	M	SD	Min	Max	Score Range
SCID-II (total)	120.4	21.3	88	173	0–224
Avoidant	11.4	2.7	7	16	0–16
Dependent	9.9	2.1	8	15	0–16
Obsessive–compulsive	11.6	3.7	8	20	0–20
Paranoid	8.1	1.4	7	11	0–12
Schizotypal	9.7	1.3	9	14	0–15
Schizoid	7.4	1	7	11	0–12
Histrionic	8.6	0.9	8	11	0–12
Narcissistic	9.9	2.2	9	18	0–20
Borderline	11.9	4.3	9	27	0–30
Antisocial	15.4	1.6	15	22	0–24
CTQ	38.2	20	25	112	25–125
AIDA	58.7	9.7	44	80	33–123
EDI	296.5	57.1	220	425	0–546

Performance (raw scores) in psychological background tasks. Min = minimum; max = maximum.

**Table 2 jcm-14-05438-t002:** Significant associations between SCID-II and GM in 18 patients with acute AN.

	Cluster Size (Number of Significant Voxels)	MNI Coordinates			t Value	*p* Value Corrected at Cluster Level (FWE)	Height Threshold
		x	y	z			
Significant correlations between SCID-II and GM at time point 1 (baseline)							
SCID total							
Mid-cingulate	770	5	18	36	5.2	0.003	0.01
Left superior parieto-occipital	715	−26	−69	39	6.3	0.005	
Histrionic							
Thalamus	845	−17	−26	5	6.5	0.002	0.01
	666	15	−20	9	4.7	0.009	
Left superior parieto-occipital	905	−30	−66	54	5.9	0.001	
Borderline							
Thalamus	1149	0	−21	9	4.8	<0.001	0.01
		−5	−11	12	4.3		
Antisocial							
Thalamus	1227	−9	−21	5	5.1	<0.001	0.01
		14	−20	6	4.4		
Significant correlations between SCID-II and GM at time point 2 (follow-up)							
SCID total							
Mid-cingulate	1062	6	−18	48	6.1	<0.001	0.01
		−9	2	51	4		
Left superior parieto-occipital	657	−27	−60	48	4.2	0.01	0.01
Histrionic							
Mid-cingulate	1111	3	−18	44	8.8	<0.001	0.01
		−3	−6	48	6.1		
Thalamus	6478	−5	−11	15	7.1	0.001	
Left superior parieto-occipital	1165	−30	−68	54	7.7	<0.001	
Right angular gyrus	909	42	−74	36	6.2	0.001	
Borderline							
Right mid-cingulate	570	14	−21	45	5.4	0.027	0.01
Antisocial							
Right mid-cingulate	501	5	−20	42	4.4	0.05	0.01

## Data Availability

The datasets used and analyzed in the present paper can be made available on request by the corresponding author due to privacy/ethical restrictions.

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
