# Peer review of "Mapping the Mind: Gray Matter Signatures of Personality Pathology in Female Adolescent Anorexia Nervosa Persist Through Treatment"

_jcm, 2025, doi:10.3390/jcm14155438_

Round 1

Reviewer 1 Report

Comments and Suggestions for Authors

General comments:
Despite being a small sample, the manuscript represents a highly relevant approach to scientific knowledge and uses methods and instruments of robust analytical consistency, capable of measuring what is proposed at the core of the manuscript, with regard to the theme of personality disorder, gray matter and anorexia nervosa in female adolescents.

Introduction
It is necessary to better describe what Anorexia Nervosa is according to the DSM.

Methods:
It is necessary to better describe the period in which the study was conducted.
It does not describe well how it was conducted and the time between the weight assessment or the recovery of body weight before and after treatment. It is important to highlight:
Was the BMI classified based on WHO parameters? Therefore, it is important to highlight this, mention the cutoff points and the reference to help the reader understand.

Lines 121-122.
In accordance with the objective of the manuscript and the methodology, the text indicates that the treatment consisted of psychoeducation, psychotherapy and family therapy. Nutritional therapy included a daily caloric intake of 2,400 to 3,000 kcal/day, monitored daily. However, how this was conducted and worked is not well described.

Abbreviations
Excellent description of abbreviations, aiding the reader's understanding.

Discussion and conclusions
Despite the very timid citation on the role of nutrition, the limitations and strengths of the study are well outlined.

References:
Although some references are current, most of them date back to periods greater than 10 or 15 years. Was the use of older references due to a lack of foundation and more recent studies? If they are really necessary, justify their use; otherwise, add other more current citations, if any, to give more robustness to the manuscript.

Author Response

General comments:
Despite being a small sample, the manuscript represents a highly relevant approach to scientific knowledge and uses methods and instruments of robust analytical consistency, capable of measuring what is proposed at the core of the manuscript, with regard to the theme of personality disorder, gray matter and anorexia nervosa in female adolescents.

Introduction
It is necessary to better describe what Anorexia Nervosa is according to the DSM.

Answer: We have provided a description of anorexia nervosa in accordance with the DSM criteria in lines 48–55.

Methods:
It is necessary to better describe the period in which the study was conducted.  It does not describe well how it was conducted and the time between the weight assessment or the recovery of body weight before and after treatment. It is important to highlight:
Was the BMI classified based on WHO parameters? Therefore, it is important to highlight this, mention the cutoff points and the reference to help the reader understand.

Answer: Thank you for your helpful comment. We added the following information. We recruited them from 2015-2017. Furthermore, we added that the weight assessment took place before the first MRI scan and after last MRI scan. We would like to clarify that since all participants in our study were between 14 and 18 years of age, BMI classification was based on BMI-for-age percentiles rather than adult BMI thresholds. Specifically, we followed the WHO growth reference for individuals aged 5–19 years. According to these guidelines, a BMI-for-age below the 5th percentile is considered indicative of significantly low weight. Accordingly, in our study, stable weight recovery was defined as achieving a BMI-for-age at or above the 5th percentile, reflecting adequate weight gain relative to age and developmental stage. We have revised the manuscript to include this clarification, along with the relevant cutoff criteria and reference, to ensure better understanding for the reader. Since all participants were between 14 and 18 years of age, BMI was classified using WHO BMI-for-age percentiles for 5- to 19-year-olds, where significantly low weight is defined as a BMI-for-age less than the 5th percentile (Engelhardt et al., 2021). In this context, stable weight recovery was defined as reaching a BMI-for-age at or above the 5th percentile, indicating sufficient weight gain for the given age and developmental stage (see lines 126-149).

Lines 121-122.
In accordance with the objective of the manuscript and the methodology, the text indicates that the treatment consisted of psychoeducation, psychotherapy and family therapy. Nutritional therapy included a daily caloric intake of 2,400 to 3,000 kcal/day, monitored daily. However, how this was conducted and worked is not well described.

We have added a paragraph on how nutritional treatment works in our clinic. This includes an individualized meal plan based on age-appropriate needs, supervised meals to ensure compliance, and regular weight monitoring. Psychoeducation and family involvement are essential components, and the approach is closely coordinated with the multidisciplinary treatment team (see participants section, lines 139-144).

Abbreviations
Excellent description of abbreviations, aiding the reader's understanding.

Discussion and conclusions
Despite the very timid citation on the role of nutrition, the limitations and strengths of the study are well outlined.

References:
Although some references are current, most of them date back to periods greater than 10 or 15 years. Was the use of older references due to a lack of foundation and more recent studies? If they are really necessary, justify their use; otherwise, add other more current citations, if any, to give more robustness to the manuscript.

We appreciate the reviewer’s thoughtful observation regarding the age of some references cited in our manuscript. Indeed, we acknowledge that a number of foundational studies—particularly in the domain of adolescent anorexia nervosa—rely on data that were collected more than a decade ago. This is largely reflective of the field itself, where long-term and longitudinal studies remain critical to understanding neurodevelopmental trajectories in this population. That said, we fully agree with the reviewer that integrating more recent findings enhances the robustness and timeliness of the work. In response, we have revised the manuscript to include several up-to-date studies from 2025 that further support and extend our conclusions. Specifically, we now reference Wan et al. (2025), who present novel findings on shared and distinct gray matter alterations in anorexia nervosa and obsessive-compulsive disorder (introduction). In line with our emphasis on the relevance of co-occurring psychopathology, we have cited Fortea et al. (2025), whose large-scale meta-analysis underscores the importance of modeling comorbidity to clarify disorder-specific neurobiological signatures (discussion). Furthermore, we incorporate findings from Kovoor et al. (2025), who provide new evidence on the dissociation between symptom severity and gray matter volume in adolescents with first-onset anorexia nervosa (introduction). These additions strengthen the manuscript’s empirical foundation and ensure a more current representation of the neurobiological literature in eating disorders

Reviewer 2 Report

Comments and Suggestions for Authors

I consider your article to be highly relevant and original. Below are some suggestions that may help improve the manuscript's quality further:

  1. The text does not mention the two subtypes of anorexia nervosa: restrictive and purging. I believe it is important to assess subtypes because, although they are different manifestations of the same psychopathology, they are usually associated with different personality traits. For example, the restrictive subtype is associated with obsessive-compulsive and avoidant personality disorders, which are Cluster C personality disorders. Conversely, the purging subtype is associated with Cluster B personality disorders, such as borderline personality disorder and narcissistic personality disorder. If this assessment has not been conducted, I suggest presenting an MRI assessment for each subtype in the future. Additionally, I suggest that the authors discuss the implications of not assessing AN subtypes in light of the results.
  2. References are missing for the statements in lines 76-78.
  3. The authors diagnosed personality disorders based on clinical interviews conducted by clinical psychologists using the Structured Clinical Interview for DSM Disorders (SCID). The average age of the participants was 15.9 years. This may be problematic. The literature expresses several concerns about diagnosing personality disorders in adolescents whose personalities are still developing. Additionally, severe malnutrition can negatively affect personality, exacerbating the maladaptive patterns typical of personality disorders. I suggest that the authors add one or two paragraphs discussing these issues.
  4. In line 109, the authors introduce the concept of "extremely underweight." I recommend that they define this term.
  5. I also recommend operationalizing the concept of nutritional restructuring. BMI for age greater than the 5th percentile alone is unreliable, as the literature indicates that this assessment should be made on a case-by-case basis, considering the weight curve prior to severe weight loss due to anorexia nervosa. For example, a young woman with a BMI slightly above the 5th percentile for her age could be malnourished and amenorrheic. Perhaps this should be included as a limitation of the study.
  6. The authors found that seven participants (39%) had a personality disorder diagnosis. Is this prevalence close to that found in previous studies? From an ethical standpoint, was the diagnosis communicated to the adolescents and their caregivers or parents? What about the health team at the inpatient unit? Were any referrals made to mental health services?
  7. I suggest that the authors include the possible score range for each instrument in Table 1. This would facilitate interpretation of the results.
  8. In lines 296–298, the authors make an important point about structural changes in the brains of the evaluated adolescents that persist even after nutritional restructuring. However, I don't believe the clinical, etiological, and pathophysiological implications of this result are clear. I noticed that you addressed this in the study's conclusion. I suggest removing it from there and incorporating it into the discussion with further expansion. Congratulations on the quality and relevance of the article. Best regards.

Author Response

I consider your article to be highly relevant and original. Below are some suggestions that may help improve the manuscript's quality further:

  1. The text does not mention the two subtypes of anorexia nervosa: restrictive and purging. I believe it is important to assess subtypes because, although they are different manifestations of the same psychopathology, they are usually associated with different personality traits. For example, the restrictive subtype is associated with obsessive-compulsive and avoidant personality disorders, which are Cluster C personality disorders. Conversely, the purging subtype is associated with Cluster B personality disorders, such as borderline personality disorder and narcissistic personality disorder. If this assessment has not been conducted, I suggest presenting an MRI assessment for each subtype in the future. Additionally, I suggest that the authors discuss the implications of not assessing AN subtypes in light of the results.

Answer: We outline that we assessed AN by using the SCID-I interview. Furthermore, we state that in our sample we included only adolescents with AN-restrictive type (see participants section)

  1. References are missing for the statements in lines 76-78.

Answer: Thank you for pointing this out. We added the following references:

Kappou K, Ntougia M, Kourtesi A, Panagouli E, Vlachopapadopoulou E, Michalacos S, Gonidakis F, Mastorakos G, Psaltopoulou T, Tsolia M, Bacopoulou F, Sergentanis TN, Tsitsika A. Neuroimaging Findings in Adolescents and Young Adults with Anorexia Nervosa: A Systematic Review. Children (Basel). 2021 Feb 12;8(2):137. doi: 10.3390/children8020137. PMID: 33673193; PMCID: PMC7918703.

Martin Monzon, B.; Henderson, L.A.; Madden, S.; Macefield, V.G.; Touyz, S.; Kohn, M.R.; Clarke, S.; Foroughi, N.; Hay, P. Grey matter volume in adolescents with anorexia nervosa and associated eating disorder symptoms. Eur J Neurosci 2017, 46, 2297-2307, doi:10.1111/ejn.13659.

  1. The authors diagnosed personality disorders based on clinical interviews conducted by clinical psychologists using the Structured Clinical Interview for DSM Disorders (SCID). The average age of the participants was 15.9 years. This may be problematic. The literature expresses several concerns about diagnosing personality disorders in adolescents whose personalities are still developing. Additionally, severe malnutrition can negatively affect personality, exacerbating the maladaptive patterns typical of personality disorders. I suggest that the authors add one or two paragraphs discussing these issues.

Answer: We added a paragraph in the discussion section on why PDs can and should be classified in adolescents; Furthermore, we gave some arguments why we should be cautious on the aspect of malnutrition as it can be linked to dimensions of personality pathology (i.e. interpersonal functioning, see lines 413-428)

  1. In line 109, the authors introduce the concept of "extremely underweight." I recommend that they define this term.

Answer: We are happy to follow this recommendation: Extremely underweight AN patients (BMI for age < 5th percentile),

  1. I also recommend operationalizing the concept of nutritional restructuring. BMI for age greater than the 5th percentile alone is unreliable, as the literature indicates that this assessment should be made on a case-by-case basis, considering the weight curve prior to severe weight loss due to anorexia nervosa. For example, a young woman with a BMI slightly above the 5th percentile for her age could be malnourished and amenorrheic. Perhaps this should be included as a limitation of the study.

Answer: Thank you very much for this important comment. We fully agree that nutritional restructuring and recovery in adolescents with AN is a complex. Individualized process that cannot be fully captured by a single BMI-for-age threshold. While we used a BMI-for-age ≥ 5th percentile as a pragmatic operational definition of stable weight recovery – consistent with WHO growth standards and common research practice – we acknowledge that this cutoff does not account for individual weight trajectories or pre-morbid growth curves. As rightly noted, adolescents who fall just above the 5th percentile may still be clinically undernourished, showing persistent symptoms such as amenorrhea or delayed pubertal development. We have therefore added this as a limitation in the manuscript: While stable weight recovery was defined as achieving a BMI-for-age at or above the 5th percentile, this cutoff may not fully capture individual differences in nutritional status, as it does not account for pre-morbid weight trajectories or physiological markers of malnutrition (lines 409-412).

  1. The authors found that seven participants (39%) had a personality disorder diagnosis. Is this prevalence close to that found in previous studies? From an ethical standpoint, was the diagnosis communicated to the adolescents and their caregivers or parents? What about the health team at the inpatient unit? Were any referrals made to mental health services?

We added a paragraph on prevalence rates consistent with findings from other studies in the section on neuropsychological background tasks (lines 269-272). Additionally, we included information regarding the communication of a PD diagnosis to parents and the clinical team in the participants section (lines 133-137).

  1. I suggest that the authors include the possible score range for each instrument in Table 1. This would facilitate interpretation of the results.

Answer: We added the score ranges for each instrument in Table 1.

  1. In lines 296–298, the authors make an important point about structural changes in the brains of the evaluated adolescents that persist even after nutritional restructuring. However, I don't believe the clinical, etiological, and pathophysiological implications of this result are clear. I noticed that you addressed this in the study's conclusion. I suggest removing it from there and incorporating it into the discussion with further expansion. Congratulations on the quality and relevance of the article. Best regards.

Answer: We removed this section from conclusions and moved it into the discussion section with further expansion:

Our findings reveal that specific structural brain alterations in adolescents with acute AN persist even after nutritional restructuring. Despite improvements in BMI over the course of treatment, no statistically significant changes were observed in GM volume when corrected for BMI, suggesting that certain neuroanatomical differences may represent stable – trait-related features rather than solely consequences of malnutrition. This persistence highlights the importance of enduring neurobiological vulnerabilities underlying AN, which may contribute to disease chronicity and influence treatment response. These results also suggest the necessity to differentiate brain abnormalities based not only on BMI or eating disorder symptoms but also on predisposing factors like comorbid PD. Recognizing characteristics of PD – such as emotional dysregulation, interpersonal problem behaviors or impulsivity – may help clinicians to tailor therapeutic strategies more effectively. For instance, individuals with higher symptoms of borderline or histrionic pathology may benefit from interventions targeting affect regulation and relational dynamics. Moreover, these neurostructural patterns may act as early markers of poorer treatment response or relapse risk, supporting the use of dimensional personality assessments in early diagnostics. Integrating disorder-specific interventions for PD alongside nutritional and psychiatric care might enhance long-term outcomes by addressing both the psychological and neurobiological dimensions of AN recovery. Understanding these lasting brain changes is therefore crucial for developing comprehensive therapeutic approaches that address not only weight restoration but also the underlying neural substrates associated with PD and symptom maintenance.

Conclusions:

Our findings support the hypothesis that PD are associated with specific focal GM alterations in brain regions important for AN symptomatology and potentially related to treatment outcomes in adolescent patients. In particular, PD in female adolescent patients with AN may be specifically related to GM alterations in the thalamus, cingulate, and parieto-occipital regions, which are present during acute illness and persist after weight restoration therapy.